# Risk of fatty liver after long-term use of tamoxifen in patients with breast cancer

Jeong-Ju Yoo[1☯], Yong Seok Lim[1☯], Min Sung Kim[2], Bora Lee[3], Bo-Yeon Kim[4], Zisun Kim[5], Ji Eun Lee[6], Min Hee Lee[6], Sang Gyune Kim[1]*, Young Seok Kim[1]

1 Division of Gastroenterology and Hepatology, Department of Internal Medicine, Soonchunhyang University School of Medicine Bucheon Hospital, Bucheon, Korea, 2 Department of Internal Medicine, Soonchunhyang University Gumi Hospital, Gumi, Korea, 3 Department of Biostatistics, Graduate School of Chung-Ang University, Seoul, Republic of Korea, 4 Division of Endocrinology, Department of Internal Medicine, Soonchunhyang University School of Medicine Bucheon Hospital, Bucheon, Korea, 5 Department of Surgery, Soonchunhyang University School of Medicine Bucheon Hospital, Bucheon, Korea, 6 Department of Radiology, Soonchunhyang University School of Medicine Bucheon Hospital, Bucheon, Korea

☯ These authors contributed equally to this work.
* mcnulty@schmc.ac.kr

**Data Availability Statement:** All relevant data are within the manuscript and its Supporting Information files.

**Funding:** The author(s) received no specific funding for this work.

## Abstract

### Background

Few studies report the effects of tamoxifen intake and the occurrence of *de novo* fatty liver and the deterioration of existing fatty liver. The aim of this study was to investigate the effects of tamoxifen on fatty change of liver over time and also the impact of fatty liver on the prognosis of patients with breast cancer.

### Methods

This was a single-center, retrospective study of patients who were diagnosed with primary breast cancer from January 2007 to July 2017. 911 consecutive patients were classified into three groups according to treatment method: tamoxifen group, aromatase inhibitor (AI) group, and control group.

### Results

Median treatment duration was 49 months (interquartile range, *IQR*; 32–58) and median observational period was 85 months (*IQR*; 50–118). Long-term use of tamoxifen significantly aggravated fatty liver status compared to AI or control groups [hazard ratio (*HR*): 1.598, 95% confidence interval (*CI*): 1.173–2.177, *P* = 0.003] after adjusting other factors. When analyzed separately depending on pre-existing fatty liver at baseline, tamoxifen was involved in the development of *de novo* fatty liver [*HR*: 1.519, 95% *CI*: 1.100–2.098, *P* = 0.011) and had greater effect on fatty liver worsening (*HR*: 2.103, 95% *CI*: 1.156–3.826, *P* = 0.015). However, the progression of fatty liver did not significantly affect the mortality of breast cancer patients.

**Competing interests:** The authors have declared that no competing interests exist.

**Abbreviations:** AI, aromatase inhibitor; ALT, alanine aminotransferase; ANOVA, analysis of variance; AST, aspartate aminotransferase; BMI, body mass index; CI, confidence interval; CT, computed tomography; ER, estrogen receptor; HR, hazard ratio; HU, hounsfield unit; MR, magnetic resonance; PR, progesterone receptor; PS, propensity score; ROI, regions of interest; SERM, selective estrogen receptor modulator; USG, ultrasonography.

## Conclusions

Tamoxifen had a significant effect on the fatty liver status compared to other treatment modalities in breast cancer patients. Although fatty liver did not affect the prognosis of breast cancer, meticulous attention to cardiovascular disease or other metabolic disease should be paid when used for a long time.

## Introduction

Breast cancer is the most common cancer in women, with an annual incidence of two million cases worldwide. Also, about 70% of breast cancer patients are hormone receptor positive.[1] Tamoxifen, a selective estrogen receptor modulator (SERM), is a well-known adjuvant endocrine treatment for estrogen receptor (ER)-positive breast cancer patients. Since tamoxifen can increase the rates of disease-free survival and overall survival of patients at all stages, and reduce the local recurrence rate, its usage in ER-positive patients is recommended for at least five years.[2, 3] However, side effects due to long-term use of tamoxifen such as vaginal bleeding, endometrial cancer, deep vein thrombosis, pulmonary embolism, and fatty liver have been reported.[4] Among these side effects, fatty liver is a serious complication because it can reduce drug compliance and increase the incidence of other metabolic diseases.[5, 6] In addition, it was reported that 43% of breast cancer patients treated with tamoxifen developed hepatic steatosis within the first two years.[7]

Aromatase inhibitor (AI) has the same mechanism as tamoxifen by inhibiting estrogen action. However, unlike tamoxifen, AI does not significantly increase the incidence of fatty liver. Recently, studies have reported the benefits of using tamoxifen for up to 10 years.[8] Thus, more attention should be paid to side effects of long-term use of tamoxifen. However, existing studies usually have short-term follow-up with the diagnosis of fatty liver based on elevated liver enzymes rather than imaging diagnosis.[9] In addition, reports on the use of tamoxifen in high risk groups of fatty liver are limited. Furthermore, the effect of fatty liver on the prognosis of breast cancer patients is not well known yet.

Thus, the purpose of this study was to investigate the effects of tamoxifen on the risk of fatty liver in comparison with other treatment modalities practicing long-term follow-ups and identify high risk groups. In addition, we investigated the effect of fatty liver caused by tamoxifen on the mortality of breast cancer patients.

## Materials and methods

### Patients and study design

This was a single-center, retrospective cohort study. We retrospectively reviewed data of patients who were first pathologically diagnosed with primary breast cancer and treated at a tertiary referral hospital from January 2007 to July 2017 (S1 Fig). Exclusion criteria were: (1) treatment period of less than two years, (2) evidence of viral hepatitis (e.g., hepatitis B surface antigen positive or hepatitis C antibody positive) or liver cirrhosis, (3) significant alcohol consumption, (4) no imaging study during the follow-up period, and (5) evidence of double primary cancer. Ongoing or recent alcohol consumption >21 standard drinks on average per week in men and >14 standard drinks on average per week in women was the criteria for significant alcohol consumption.[10]

911 patients were enrolled and classified into three groups according to treatment method: 1) tamoxifen group, patients taking tamoxifen for more than two years; 2) aromatase inhibitor group, patients taking anastrozole or letrozole for more than 2 years; and 3) control group, patients who had received treatment other than tamoxifen or AI (*e.g.*, conventional chemotherapy, radiotherapy, or surgery alone). The chemotherapeutic agents of anticancer drugs were diverse, however, anthracycline and taxane-based chemotherapy was most widely used. Clinical, histological, imaging, and laboratory records of these patients were retrospectively reviewed. Patients were prescribed tamoxifen or AI every three months with laboratory tests at each regular visit. To monitor possible recurrences of breast cancer, follow-up computed tomography (CT) or ultrasonography (USG) were performed every 6months for patients with high risk of recurrence, and every 12months for others as recommended in the NCCN guideline.[11] If recurrence was not observed for five years, monitoring was carried out on a yearly basis.

Informed consent was waived because of the retrospective nature of the study and the analysis used anonymous clinical data. This study was approved by the Institutional Review Board of Soonchunhyang University Hospital, Bucheon, Korea (SCHBC-2018-12-002-001). The study protocol conformed to the ethical guidelines of the World Medical Association Declaration of Helsinki.

## Diagnosis of fatty liver

The incidence or severity of fatty liver was assessed by non-contrast CT or abdominal USG. We used only one technique (among CT or USG) as a diagnostic tool to determine fatty liver throughout the follow-up period. All the images were evaluated by two experienced radiologists (LMH and LJE) who were blinded to the study aims and the administered drug. In USG examination, the severity of fatty liver was graded as normal, mild, moderate, or severe according to echogenicity of the liver parenchyma.[12] In non-contrast CT examination, hepatic steatosis was assessed based on Hounsfield unit (HU) hepatic attenuation using published method.[13, 14] Hepatic attenuation was measured as mean of three circular regions of interest (ROIs) on three transverse sections at different hepatic levels: confluence of right hepatic vein, umbilical portion of left portal vein, and posterior branch of the right portal vein. Mean splenic attenuation was also calculated for three random area ROI values of attenuation measurement at three different splenic levels. Fatty liver was defined when the ratio of mean hepatic attenuation to mean splenic attenuation was lower than 0.9.[13, 14]

## Definition

Baseline was regarded as just before receiving hormonal therapy after breast cancer surgery and follow-up CT scan. The follow-up period was considered the period either during hormonal drug intake or until the time of loss or death. Patients were classified into the following four groups according to existence of fatty liver at baseline and the change of fatty liver status during follow-up period. *No fatty liver group* was defined as having a normal liver at baseline and remaining normal during follow-up period. *De novo fatty liver group* was defined as having a normal liver at baseline, but developing a new fatty liver. *Stable fatty liver group* was defined as having a fatty liver at baseline and no change of fatty liver severity during follow-up period. *Worsened fatty liver group* was defined as having a fatty liver at baseline and worsening of fatty liver status at any point during follow-up period (*e.g.*, increased grade of fatty liver on abdominal USG or decreasing liver attenuation index value on non-contrast CT).[15, 16] These four groups mentioned above were also categorized based on aggravation of fatty liver.

*De novo* fatty liver group and worsened fatty liver group were categorized as *progression group* while no fatty liver group and stable fatty liver group categorized as *non-progression group*.

## Statistical analysis

Frequencies and percentages were used for descriptive statistics. Statistical differences between groups were investigated using one-way analysis of variance (ANOVA) or Student's t-test for continuous variables and chi-squares test or Fisher's exact test for categorical variables as appropriate. Kaplan-Meier survival analysis was used for survival rate and incidence of fatty liver. To identify predictive factors associated with development of fatty liver according to age, Cox proportional hazards (PH) regression analysis was used. Any variable showing a significance at 0.1 in the univariate model was selected as a candidate for the multivariable model. [17] The final multiple Cox PH regression model was chosen by the stepwise selection based on the Akaike information criterion (*AIC*).

Propensity score (PS) matching analysis was conducted to minimize the probability of selection bias by pairing tamoxifen group and control group based on propensity scores. PS matching was generated by multiple logistic regressions. This model included all variables such as age, body mass index (BMI), breast cancer stage, diabetes, and hypertension that are likely to affect the process of fatty deposition in the liver. We used the nearest available matching (1:1) method for PS matching with the caliper of 0.05. For validation, we generated additional matched datasets using the caliper of 0.03 and 0.01.

All statistical analyses were performed using R (version 3.6.1, The R Foundation for Statistical Computing, Vienna, Austria) and SPSS software (version 21.0; SPSS Inc., Chicago, IL, USA). Statistical significance was defined at $P < 0.05$.

## Results

### Patients' demographics at baseline

Baseline characteristics of 911 patients enrolled in this study are described in Table 1. Tamoxifen, prescribed to 416 (45.7%) patients, accounted for the highest proportion followed by a group of patients (296, 32.5%) who did not take any anti-estrogen hormonal treatment, and next by a group of patients (199, 21.8%) who were prescribed AI. Mean age and BMI were 50.13 ± 10.32 years and 24.17 ± 3.62, respectively. Median treatment duration was 49 months [interquartile range (*IQR*) 32–58)], and median observational period was 85 months (*IQR* 50–118). From the outset 260 (28.5%) of patients already had fatty liver. Seventy (7.7%) patients had diabetes and 176 (19.3%) had hypertension. Mean follicle stimulating hormone (FSH) level was 37.38 ± 30.15 IU/L. Breast cancer of stage I was most common (42.4%), followed by stage II (40.8%) and stage III (8.1%). Lymph node metastasis was found in 284 (31.2%) patients.

### Changes in fatty liver status over time

There was no change of fatty liver status in 499 (54.8%) patients (non-progression group) while there was a deterioration of fatty liver status in 412 (45.2%) patients (progression group). Table 2 shows the baseline data according to the change of fatty liver status. BMI, aspartate aminotransferase (AST), alanine aminotransferase (ALT), and fasting blood glucose of patients in the progression group were significantly higher than those in the non-progression group. The worsened fatty liver group had the highest mean BMI, followed by stable fatty liver, *de novo* fatty liver, and no fatty liver groups. Levels of liver enzymes including AST, ALT, and other lipid profiles showed the same pattern. Of note, tamoxifen significantly increased fatty

**Table 1. Baseline characteristics of patients.**

| Variable | All | Control | Tamoxifen | Aromatase inhibitor | P-value |
|---|---|---|---|---|---|
| | (N = 911) | (N = 296) | (N = 416) | (N = 199) | |
| Age (year) | 50.13 ± 10.32 | 49.78 ± 10.60 | 46.87 ± 9.43 | 57.46 ± 7.68 | <0.001 |
| Body mass index kg/m$^2$) | 24.17 ± 3.62 | 24.18 ± 3.78 | 23.67 ± 3.32 | 25.22 ± 3.76 | <0.001 |
| Diabetes mellitus | 70 (7.68%) | 21 (7.09%) | 24 (5.77%) | 25 (12.56%) | 0.011 |
| Hypertension | 176 (19.32%) | 49 (16.55%) | 72 (17.31%) | 55 (27.64%) | 0.003 |
| Treatment duration (month) | 49.04 ± 21.85 | 50.88 ± 30.87 | 50.00 ± 17.07 | 44.30 ± 11.55 | 0.005 |
| Cancer-related factor | | | | | |
| Stage | | | | | <0.001 |
| 0 | 71 (7.79%) | 19 (6.42%) | 49 (11.78%) | 3 (1.51%) | |
| 1 | 386 (42.37%) | 129 (43.58%) | 160 (38.46%) | 97 (48.74%) | |
| 2 | 372 (40.83%) | 120 (40.54%) | 167 (40.14%) | 85 (42.71%) | |
| 3 | 74 (8.12%) | 22 (7.43%) | 38 (9.13%) | 14 (7.04%) | |
| 4 | 8 (0.88%) | 6 (2.03%) | 2 (0.48%) | 0 (0%) | |
| Pathology | | | | | 0.007 |
| Invasive ductal carcinoma | 757 (83.10%) | 248 (83.78%) | 332 (79.81%) | 177 (88.94%) | |
| Ductal carcinoma in situ | 73 (8.01%) | 22 (7.43%) | 48 (11.54%) | 3 (1.51%) | |
| Mucinous carcinoma | 24 (2.63%) | 4 (1.35%) | 12 (2.88%) | 8 (4.02%) | |
| Infiltrating lobular carcinoma | 27 (2.96%) | 9 (3.04%) | 12 (2.88%) | 6 (3.02%) | |
| Intraductal papilloma | 13 (1.43%) | 2 (0.68%) | 6 (1.44%) | 5 (2.51%) | |
| Tubular carcinoma | 2 (0.22%) | 1 (0.34%) | 1 (0.24%) | 0 (0%) | |
| Apocrine carcinoma | 2 (0.22%) | 1 (0.34%) | 1 (0.24%) | 0 (0%) | |
| Squamous carcinoma | 2 (0.22%) | 1 (0.34%) | 1 (0.24%) | 0 (0%) | |
| Medullary carcinoma | 5 (0.55%) | 3 (1.01%) | 2 (0.48%) | 0 (0%) | |
| Others | 6 (0.66%) | 5 (1.69%) | 1 (0.24%) | 0 (0%) | |
| Lymph node metastasis | 284 (31.17%) | 107 (36.15%) | 115 (27.64%) | 62 (31.16%) | 0.054 |
| ER (Intermediate or High) | 614 (67.70%) | 108 (36.49%) | 328 (79.61%) | 178 (89.45%) | <0.001 |
| PR (Intermediate or High) | 555 (61.19%) | 102 (34.46%) | 313 (75.97%) | 140 (70.35%) | <0.001 |
| HER2 (Intermediate or High) | 340 (37.32%) | 83 (28.04%) | 167 (40.14%) | 90 (45.23%) | <0.001 |
| p53 | 328 (41.41%) | 121 (49.79%) | 152 (41.30%) | 55 (30.39%) | <0.001 |
| Ki67 (≥ 40%) | 124 (15.31%) | 77 (29.96%) | 31 (8.36%) | 16 (8.79%) | <0.001 |
| Laboratory test | | | | | |
| FSH (IU/L) | 37.38 ± 30.15 | 30.37 ± 29.13 | 34.34 ± 29.78 | 55.55 ± 25.18 | <0.001 |
| Platelet ($10^3$ mm$^3$) | 238.66 ± 72.70 | 247.55 ± 66.89 | 231.54 ± 74.33 | 240.35 ± 76.27 | 0.007 |
| AST (U/L) | 23.85 ± 13.53 | 21.28 ± 9.07 | 25.60 ± 15.42 | 24.03 ± 14.28 | <0.001 |
| ALT (U/L) | 22.17 ± 16.86 | 19.35 ± 15.97 | 23.93 ± 18.43 | 22.67 ± 13.98 | <0.001 |
| Serum albumin (mg/dL) | 4.20 ± 0.45 | 4.36 ± 0.45 | 4.11 ± 0.43 | 4.14 ± 0.43 | <0.001 |
| Total bilirubin (mg/dL) | 0.59 ± 0.33 | 0.65 ± 0.42 | 0.55 ± 0.25 | 0.58 ± 0.31 | <0.001 |
| Total cholesterol (mg/dL) | 185.16 ± 35.97 | 185.90 ± 36.59 | 181.12 ± 33.83 | 192.48 ± 38.28 | 0.118 |
| Triglyceride (mg/dL) | 132.95 ± 96.18 | 146.31 ± 110.62 | 124.34 ± 91.42 | 133.36 ± 82.19 | 0.060 |
| HDL-cholesterol (mg/dL) | 53.40 ± 13.41 | 52.29 ± 13.02 | 55.45 ± 13.92 | 51.18 ± 12.51 | 0.002 |
| LDL-cholesterol (mg/dL) | 107.32 ± 32.39 | 107.98 ± 31.99 | 104.32 ± 30.80 | 112.13 ± 35.42 | 0.096 |
| Fasting blood glucose (mg/dL) | 110.07 ± 28.42 | 106.83 ± 24.72 | 109.74 ± 28.28 | 115.60 ± 32.83 | 0.001 |

computed by one-way ANOVA for continuous variables and chi-squares test for categorical variables.

Abbreviations: ER, estrogen receptor; PR, progesterone receptor; HER2, Human epidermal growth factor receptor 2; FSH, follicle stimulating hormone; AST, aspartate aminotransferase; ALT, alanine aminotransferase; HDL, high-density lipoprotein; LDL, low-density lipoprotein; NFS, nonalcoholic fatty liver disease fibrosis score; FIB-4, fibrosis-4; CT, computed tomography.

**Table 2. Clinical characteristics depending on the status of fatty liver.**

| Variable | All | No fatty liver | *De novo* fatty liver | Stable fatty liver | Worsened fatty liver | P | Fatty liver progression* (-) | Fatty liver progression* (+) | P |
|---|---|---|---|---|---|---|---|---|---|
| | (N = 911) | (N = 364) | (N = 287) | (N = 135) | (N = 125) | | (N = 499) | (N = 412) | |
| Treatment modality | | | | | | <0.001 | | | <0.001 |
| Control | 296 | 159 (53.72%) | 67 (22.64%) | 52 (17.57%) | 18 (6.07%) | | 211 (71.29%) | 85 (28.71%) | |
| Aromatase inhibitor | 199 | 73 (36.68%) | 55 (27.64%) | 41 (20.60%) | 30 (15.08%) | | 114 (57.28%) | 85 (42.72%) | |
| Tamoxifen | 416 | 132 (31.73%) | 165 (39.66%) | 42 (10.10%) | 77 (18.51%) | | 174 (41.83%) | 242 (58.17%) | |
| Age (year) | 50.13 ± 10.32 | 49.65 ± 10.69 | 49.70 ± 10.40 | 52.58 ± 10.30 | 49.84 ± 8.64 | 0.018 | 50.44 ± 10.66 | 49.75 ± 9.89 | 0.306 |
| Body mass index (kg/m²) | 24.17 ± 3.62 | 23.03 ± 3.15 | 24.06 ± 3.25 | 25.38 ± 3.92 | 26.47 ± 3.93 | <0.001 | 23.67 ± 3.53 | 24.79 ± 3.64 | <0.001 |
| Diabetes mellitus | 70 (7.68%) | 18 (4.95%) | 18 (6.27%) | 18 (13.33%) | 16 (12.80%) | 0.002 | 36 (7.21%) | 34 (8.25%) | 0.645 |
| Hypertension | 176 (19.32%) | 55 (15.11%) | 52 (18.12%) | 33 (24.44%) | 36 (28.80%) | 0.003 | 88 (17.64%) | 88 (21.36%) | 0.183 |
| Treatment duration (month) | 49.04 ± 21.85 | 44.48 ± 20.83 | 48.57 ± 18.57 | 60.01 ± 28.09 | 51.55 ± 19.64 | <0.001 | 48.68 ± 24.01 | 49.48 ± 18.92 | 0.075 |
| Cancer-related factor | | | | | | | | | |
| Stage | | | | | | 0.460 | | | 0.062[a] |
| 0 | 71 (7.79%) | 26 (7.14%) | 27 (9.41%) | 5 (3.70%) | 13 (10.40%) | | 31 (6.21%) | 40 (9.71%) | |
| 1 | 386 (42.37%) | 171 (46.98%) | 110 (38.33%) | 60 (44.44%) | 45 (36.00%) | | 231 (46.29%) | 155 (37.62%) | |
| 2 | 372 (40.83%) | 138 (37.91%) | 121 (42.16%) | 57 (42.22%) | 56 (44.80%) | | 195 (39.08%) | 177 (42.96%) | |
| 3 | 74 (8.12%) | 26 (7.14%) | 26 (9.06%) | 12 (8.89%) | 10 (8.00%) | | 38 (7.62%) | 36 (8.74%) | |
| 4 | 8 (0.88%) | 3 (0.82%) | 3 (1.05%) | 1 (0.74%) | 1 (0.80%) | | 4 (0.80%) | 4 (0.97%) | |
| Pathology | | | | | | 0.099 | | | |
| Invasive ductal carcinoma | 757 (83.10%) | 312 (85.71%) | 229 (79.79%) | 117 (86.67%) | 99 (79.20%) | | | | 0.101 |
| Ductal carcinoma in situ | 73 (8.01%) | 27 (7.42%) | 26 (9.06%) | 6 (4.44%) | 14 (11.20%) | | 429 (85.97%) | 328 (79.61%) | |
| Mucinous carcinoma | 24 (2.63%) | 6 (1.65%) | 11 (3.83%) | 4 (2.96%) | 3 (2.40%) | | 33 (6.61%) | 40 (9.71%) | |
| Infiltrating lobular carcinoma | 27 (2.96%) | 13 (3.57%) | 11 (3.83%) | 3 (2.22%) | 0 (0%) | | 10 (2.00%) | 14 (3.40%) | |
| Intraductal papilloma | 13 (1.43%) | 1 (0.27%) | 7 (2.44%) | 1 (0.74%) | 4 (3.20%) | | 16 (3.21%) | 11 (2.67%) | |
| Tubular carcinoma | 2 (0.22%) | 1 (0.27%) | 0 (0%) | 0 (0%) | 1 (0.80%) | | 2 (0.40%) | 11 (2.67%) | |
| Apocrine carcinoma | 2 (0.22%) | 1 (0.27%) | 0 (0%) | 0 (0%) | 1 (0.80%) | | 1 (0.20%) | 1 (0.24%) | |
| Squamous carcinoma | 2 (0.22%) | 0 (0%) | 1 (0.35%) | 1 (0.74%) | 0 (0%) | | 1 (0.20%) | 1 (0.24%) | |
| Medullary carcinoma | 5 (0.55%) | 1 (0.27%) | 2 (0.70%) | 1 (0.74%) | 1 (0.80%) | | 1 (0.20%) | 1 (0.24%) | |
| Others | 6 (0.66%) | 2 (0.55%) | 0 (0%) | 2 (1.48%) | 2 (1.60%) | | 2 (0.40%) | 3 (0.73%) | |
| Lymph node metastasis | 284 (31.17%) | 104 (28.57%) | 96 (33.45%) | 47 (34.81%) | 37 (29.60%) | 0.420 | 151 (30.26%) | 133 (32.28%) | 0.559 |
| ER (Intermediate or High) | 614 (67.70%) | 230 (63.19%) | 208 (73.24%) | 82 (61.19%) | 94 (75.20%) | 0.004 | 312 (62.65%) | 302 (73.84%) | <0.001 |
| PR (Intermediate or High) | 555 (61.19%) | 200 (54.95%) | 192 (67.61%) | 70 (52.24%) | 93 (74.40%) | <0.001 | 270 (54.22%) | 285 (69.68%) | <0.001 |
| HER2 (Intermediate or High) | 340 (37.32%) | 134 (36.81%) | 111 (38.68%) | 39 (28.89%) | 56 (44.80%) | 0.061 | 173 (34.67%) | 167 (40.53%) | 0.080 |
| p53 | 328 (41.41%) | 135 (42.86%) | 97 (39.75%) | 55 (45.08%) | 41 (36.94%) | 0.544 | 190 (43.48%) | 138 (38.87%) | 0.216 |
| Ki67 (≥ 40%) | 124 (15.31%) | 58 (17.85%) | 33 (13.10%) | 22 (18.03%) | 11 (9.91%) | 0.124 | 80 (17.90%) | 44 (12.12%) | 0.030 |

(*Continued*)

**Table 2.** (Continued)

| Variable | All | No fatty liver | *De novo* fatty liver | Stable fatty liver | Worsened fatty liver | P | Fatty liver progression* (-) | Fatty liver progression* (+) | P |
|---|---|---|---|---|---|---|---|---|---|
| | (N = 911) | (N = 364) | (N = 287) | (N = 135) | (N = 125) | | (N = 499) | (N = 412) | |
| Laboratory test | | | | | | | | | |
| Initial | | | | | | | | | |
| FSH (IU/L) | 37.38 ± 30.15 | 37.01 ± 31.61 | 33.93 ± 28.73 | 43.40 ± 29.05 | 39.98 ± 28.98 | 0.048 | 38.65 ± 31.07 | 35.74 ± 28.89 | 0.416 |
| Platelet ($10^3$ $mm^3$) | 238.66 ± 72.70 | 237.47 ± 73.38 | 241.26 ± 74.23 | 233.10 ± 71.58 | 242.18 ± 68.66 | 0.562 | 236.29 ± 72.85 | 241.54 ± 72.51 | 0.216 |
| AST (U/L) | 23.85 ± 13.53 | 21.79 ± 9.42 | 22.43 ± 9.37 | 26.08 ± 17.47 | 30.73 ± 21.92 | <0.001 | 22.95 ± 12.27 | 24.95 ± 14.86 | 0.033 |
| ALT (U/L) | 22.17 ± 16.86 | 18.39 ± 13.70 | 20.43 ± 12.38 | 27.13 ± 21.14 | 31.80 ± 23.05 | <0.001 | 20.75 ± 16.49 | 23.88 ± 17.16 | <0.001 |
| Serum albumin (mg/dL) | 4.20 ± 0.45 | 4.22 ± 0.46 | 4.14 ± 0.47 | 4.23 ± 0.44 | 4.23 ± 0.38 | 0.092 | 4.22 ± 0.45 | 4.17 ± 0.45 | 0.040 |
| Total bilirubin (mg/dL) | 0.59 ± 0.33 | 0.60 ± 0.27 | 0.56 ± 0.27 | 0.64 ± 0.57 | 0.58 ± 0.25 | 0.081 | 0.61 ± 0.38 | 0.56 ± 0.26 | 0.024 |
| Total cholesterol (mg/dL) | 185.16 ± 35.97 | 182.03 ± 37.16 | 183.82 ± 34.51 | 193.00 ± 36.18 | 188.87 ± 34.33 | 0.006 | 185.00 ± 37.18 | 185.35 ± 34.50 | 0.884 |
| Triglyceride (mg/dL) | 132.95 ± 96.18 | 109.55 ± 64.12 | 132.42 ± 90.74 | 144.71 ± 82.52 | 185.90 ± 156.16 | <0.001 | 119.58 ± 71.52 | 148.85 ± 117.20 | <0.001 |
| HDL-cholesterol (mg/dL) | 53.40 ± 13.41 | 54.86 ± 13.91 | 54.02 ± 13.85 | 52.64 ± 12.87 | 48.47 ± 9.98 | <0.001 | 54.30 ± 13.67 | 52.36 ± 13.05 | 0.059 |
| LDL-cholesterol (mg/dL) | 107.32 ± 32.39 | 105.79 ± 31.31 | 104.31 ± 30.29 | 113.44 ± 34.13 | 113.38 ± 37.54 | 0.039 | 107.68 ± 32.16 | 106.89 ± 32.72 | 0.743 |
| Fasting blood glucose (mg/dL) | 110.07 ± 28.42 | 107.40 ± 25.31 | 109.61 ± 30.42 | 112.27 ± 31.95 | 116.58 ± 27.28 | 0.001 | 108.72 ± 27.32 | 111.72 ± 29.64 | 0.043 |
| BARD | 1.95 ± 0.77 | 1.96 ± 0.62 | 1.95 ± 0.74 | 1.93 ± 0.94 | 1.95 ± 1.01 | 0.930 | 1.95 ± 0.72 | 1.95 ± 0.83 | 0.984 |
| NFS | -1.04 ± 1.31 | -1.05 ± 1.30 | -1.06 ± 1.35 | -0.93 ± 1.30 | -1.10 ± 1.25 | 0.589 | -1.01 ± 1.30 | -1.07 ± 1.32 | 0.393 |
| FIB-4 | 1.27 ± 1.03 | 1.26 ± 0.72 | 1.24 ± 1.23 | 1.38 ± 1.43 | 1.26 ± 0.82 | 0.332 | 1.29 ± 0.96 | 1.25 ± 1.12 | 0.083 |
| ALT elevation | | | | | | <0.001 | | | 0.461 |
| No | 524 (57.52%) | 224 (61.54%) | 187 (65.16%) | 69 (51.11%) | 44 (35.20%) | | 293 (58.72%) | 231 (56.07%) | |
| Yes | 387 (42.48%) | 140 (38.46%) | 100 (34.84%) | 66 (48.89%) | 81 (64.80%) | | 206 (41.28%) | 181 (43.93%) | |

Data are reported as means and standard deviation (SD) (mean ± SD) for continuous variables and frequency (percentage) for categorical variables. Proportions are presented as percentages for categorical variables. P-values were computed by one-way ANOVA or Student's t-test for continuous variables and chi-squares test or Fisher's exact test for categorical variables as appropriate ([a], computed by Fisher's exact test).

Abbreviations: ER, estrogen receptor; PR, progesterone receptor; HER2, Human epidermal growth factor receptor 2; FSH, follicle stimulating hormone; AST, aspartate aminotransferase; ALT, alanine aminotransferase; HDL, high-density lipoprotein; LDL, low-density lipoprotein; NFS, nonalcoholic fatty liver disease fibrosis score; FIB-4, fibrosis-4.

* Definition of fatty liver progression is increased grade of fatty liver according to abdominal ultrasound or decreasing liver attenuation index value on non-contrast computed tomography.

liver progression compared to AI or control over time (*P* < 0.001, Fig 1A). These differences were more pronounced in patients who had previously fatty liver (Fig 1B). In patients without basal fatty liver, tamoxifen also increased the incidence of *de novo* fatty liver more than control (Fig 1C).

Next, we investigated factors affecting fatty liver progression by Cox proportional hazard regression analyses. Table 3 shows that tamoxifen significantly contributed to the progression of fatty liver [hazard ratio (*HR*): 1.598, 95% confidence interval (*CI*): 1.173–2.177, *P* = 0.003] as compared with other treatments modality after adjusting for BMI, progesterone receptor (PR), and FSH. This effect of tamoxifen on fatty liver was consistently meaningful whether or not fatty liver was present at baseline (S1 Table).

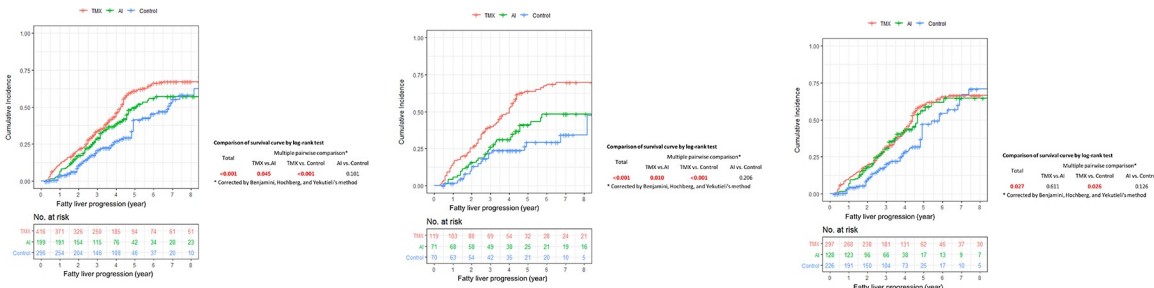

**Fig 1. Cumulative incidence of fatty liver progression before propensity score matching.** (A) All patients (N = 911), (B) Patients with initial fatty liver (N = 260), (C) Patients without initial fatty liver (N = 651).

## Propensity score matching analysis

We performed PS matching in these patients (S2 Table). After PS matching, factors that might affect the development of fatty liver such as age, BMI, cancer stage, diabetes, and hypertension were well-balanced (S2 Fig). In the matched cohort, tamoxifen significantly increased the incidence of fatty liver progression compared with the control group (P<0.001, Fig 2A). And this phenomenon was observed in the same manner regardless of fatty liver status at baseline (Fig 2B and 2C). Univariate (S3 Table) and multivariate analyses (Table 4) according to baseline fatty liver status were also performed for the matched cohort, showing similar results with non-matched cohort. Overall, tamoxifen increased the risk of fatty liver by 1.385 times (95% CI: 1.019–1.883, P = 0.037) compared to control after adjusting for BMI and PR status. The effect of tamoxifen on fatty liver progression was also confirmed to the same degree and significance in other matched datasets (HR: 1.4, 95% CI: 1.018–1.927 in dataset with caliper 0.3; HR: 1.452, 95% CI: 1.021–2.065 in dataset with caliper 0.1) (S5 and S6 Tables).

## Prediction model associated with fatty liver progression

Apart from multivariate analysis, we have made prediction models 1, 2, 3, and 4, incorporating various confounding factors to re-confirm the causal relationship between tamoxifen use and fatty liver progression. As shown in Table 5, tamoxifen was significantly associated with progression of fatty liver regardless of all these factors. After adjusting for age, BMI, diabetes, hypertension, menopausal status, cancer-related factors and other laboratory findings, tamoxifen was independently associated with increased risk of fatty liver (before matching, HR: 2.998, 95% CI: 1.646–5.462, P < 0.001; after matching, HR: 2.907, 95% CI: 1.451–5.821, P = 0.003).

Next, we performed a subgroup analysis in the tamoxifen group to find any correlations between the degree of hepatic steatosis with drug duration or changes in liver enzymes. However, we discovered no significant or clinically meaningful correlations when using the spearman's correlation coefficient (data not shown).

## Effect of fatty liver progression on mortality

In addition, we analyzed the effect of fatty liver on the prognosis of breast cancer patients. When patients were classified into two groups according to progression of fatty liver, there was no significant difference in long-term mortality between the two groups (before matching, P = 0.928, S3A Fig; after matching, P = 0.471, S3B Fig). Given the follow-up period with our cohort, cardiovascular risk possibly caused by tamoxifen was not fully assessed. In univariate (S4 Table) and multivariate analyses (Table 6), progression of fatty liver did not significantly

**Table 3. Multivariable analysis for the risk factors associated with fatty liver progression.**

| Variable | Univariable | | Multivariable | |
|---|---|---|---|---|
| | *HR (95% CI)* | **P-value** | *HR (95% CI)* | **P-value** |
| Treatment modality | | | | |
| Control | 1 (Reference) | | 1 (Reference) | |
| Aromatase inhibitor | 1.331 (0.985–1.798) | 0.063 | 1.230 (0.839–1.803) | 0.289 |
| Tamoxifen | 1.744 (1.361–2.234) | <0.001 | 1.598 (1.173–2.177) | 0.003 |
| Age (year) | 1.002 (0.993–1.012) | 0.618 | | |
| BMI (kg/m$^2$) | 1.061 (1.036–1.087) | <0.001 | 1.083 (1.052–1.114) | <0.001 |
| Diabetes | 1.197 (0.843–1.701) | 0.315 | | |
| Hypertension | 1.214 (0.959–1.537) | 0.107 | | |
| Cancer stage | | | | |
| ≤1 | 1 (Reference) | | | |
| 2 | 1.230 (1.003–1.507) | 0.047 | | |
| ≥3 | 1.204 (0.857–1.693) | 0.284 | | |
| Pathology | | | | |
| Invasive ductal carcinoma | 1 (Reference) | | | |
| Ductal carcinoma in situ | 1.271 (0.915–1.764) | 0.153 | | |
| Mucinous carcinoma | 1.300 (0.761–2.22) | 0.337 | | |
| Infiltrating lobular carcinoma | 1.051 (0.576–1.917) | 0.871 | | |
| Intraductal papilloma | 2.187 (1.199–3.991) | 0.011 | | |
| Tubular carcinoma | 0.878 (0.123–6.252) | 0.896 | | |
| Apocrine carcinoma | 0.923 (0.13–6.574) | 0.936 | | |
| Squamous carcinoma | 1.005 (0.141–7.16) | 0.996 | | |
| Medullary carcinoma | 0.900 (0.289–2.807) | 0.856 | | |
| Others | 0.820 (0.204–3.294) | 0.780 | | |
| Lymph node metastasis | 1.115 (0.907–1.37) | 0.303 | | |
| ER (Intermediate or High) | 1.459 (1.17–1.82) | 0.001 | | |
| PR (Intermediate or High) | 1.621 (1.312–2.002) | <0.001 | 1.395 (1.049–1.857) | 0.022 |
| HER2 (Intermediate or High) | 1.215 (0.998–1.479) | 0.052 | | |
| Chemotherapy | 0.827 (0.664–1.029) | 0.089 | | |
| Radiotherapy | 1.092 (0.9–1.326) | 0.372 | | |
| FSH | 0.995 (0.991–0.999) | 0.012 | 0.995 (0.991–0.999) | 0.024 |
| Platelet | 1.002 (1.001–1.003) | 0.004 | 1.002 (1.001–1.004) | 0.004 |
| AST | 1.004 (0.998–1.011) | 0.207 | | |
| ALT | 1.004 (0.999–1.009) | 0.115 | | |
| Serum albumin | 0.902 (0.729–1.115) | 0.339 | | |
| Total bilirubin | 0.620 (0.419–0.917) | 0.017 | | |
| Total cholesterol | 0.999 (0.997–1.002) | 0.668 | | |
| Triglyceride | 1.001 (1–1.002) | 0.002 | | |
| HDL-cholesterol | 0.983 (0.975–0.992) | <0.001 | | |
| LDL-cholesterol | 1 (0.996–1.003) | 0.972 | | |
| Fasting blood glucose | 1.003 (1–1.006) | 0.088 | | |
| BARD | 1.044 (0.921–1.184) | 0.503 | | |
| NFS | 0.947 (0.876–1.024) | 0.170 | | |
| FIB-4 | 0.937 (0.83–1.056) | 0.287 | | |

Abbreviations: *HR*, hazard ratio; *CI*, confidence interval; ER, estrogen receptor; PR, progesterone receptor; HER2, Human epidermal growth factor receptor 2; FSH, follicle stimulating hormone; AST, aspartate aminotransferase; ALT, alanine aminotransferase; HDL, high-density lipoprotein; LDL, low-density lipoprotein; NFS, nonalcoholic fatty liver disease fibrosis score; FIB-4, fibrosis-4.

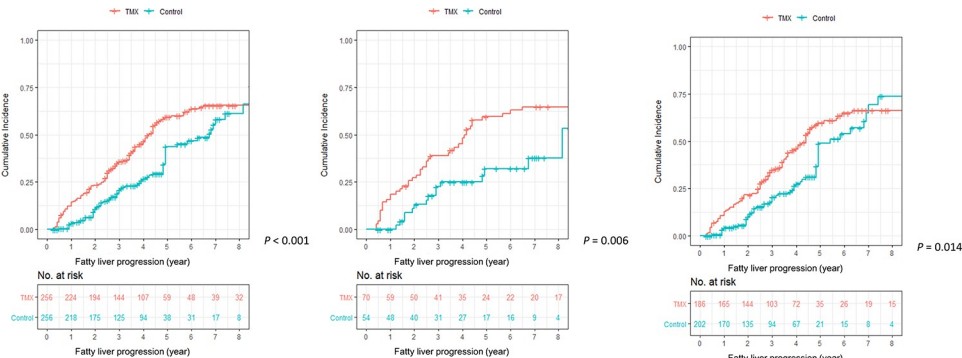

**Fig 2. Cumulative incidence of fatty liver progression after propensity score matching.** (A) All patients (N = 512), (B) Patients with initial fatty liver (N = 124), (C) Patients without initial fatty liver (N = 388).

affect mortality of breast cancer patients (*HR*: 0.753, 95% *CI*: 0.380–1.493, *P* = 0.417), although cancer stage and age were associated with mortality.

## Discussion

Recent studies have shown that 10 years of adjuvant treatment of tamoxifen is superior to 5 years of tamoxifen use in reducing the risk of breast cancer recurrence or death.[8, 18] Since then, extended adjuvant tamoxifen (10 years) is recommended for selected high risk groups. However, there are still few studies considering the effect of tamoxifen on the incidence of

**Table 4. Propensity score matching analysis for risk factors associated with fatty liver progression.**

| Variable | Multivariable | |
|---|---|---|
| | HR (95% CI) | P-value |
| **All (N = 512)** | | |
| Treatment modality | | |
| Control | 1 (Reference) | |
| Tamoxifen | 1.385 (1.019–1.883) | 0.037 |
| Body mass index (kg/m$^2$) | 1.069 (1.032–1.107) | <0.001 |
| PR (Intermediate or High) | 1.607 (1.186–2.178) | 0.002 |
| **Fatty liver (-) at baseline (N = 388)** | | |
| Treatment modality | | |
| Control | 1 (Reference) | |
| Tamoxifen | 2.214 (1.416–3.464) | <0.001 |
| Body mass index (kg/m$^2$) | 1.067 (1.014–1.122) | 0.013 |
| Triglyceride | 1.005 (1.002–1.007) | <0.001 |
| **Fatty liver (+) at baseline (N = 124)** | | |
| Treatment modality | | |
| Control | 1 (Reference) | |
| Tamoxifen | 2.103 (1.156–3.826) | 0.015 |
| Body mass index (kg/m$^2$) | 1.049 (1.005–1.096) | 0.029 |
| HER2 (Intermediate + High) | 1.804 (1.06–3.071) | 0.030 |
| Radiotherapy | 2.081 (1.217–3.56) | 0.007 |
| Total cholesterol | 0.988 (0.981–0.996) | 0.002 |

Abbreviations: *HR*, hazard ratio; *CI*, confidence interval.

**Table 5. Prediction model related to fatty liver progression.**

| Variable | | N | Model 1 [*] | | Model 2 [†] | | Model 3 [§] | | Model 4 [¶] | |
|---|---|---|---|---|---|---|---|---|---|---|
| | | | HR (95% CI) | P | HR (95% CI) | P | HR (95% CI) | P | HR (95% CI) | P |
| **Before matching** | **All** | | | | | | | | | |
| | Control | 296 | 1 (Reference) | | 1 (Reference) | | 1 (Reference) | | 1 (Reference) | |
| | Aromatase inhibitor | 199 | 1.331 (0.985–1.798) | 0.063 | 1.232 (0.902–1.682) | 0.190 | 1.191 (0.858–1.654) | 0.295 | 1.381 (0.806–2.367) | 0.239 |
| | Tamoxifen | 416 | 1.744 (1.361–2.234) | <0.001 | 1.882 (1.461–2.424) | <0.001 | 1.860 (1.416–2.444) | <0.001 | 1.859 (1.167–2.96) | 0.009 |
| | **Fatty liver (-) at baseline** | | | | | | | | | |
| | Control | 226 | 1 (Reference) | | 1 (Reference) | | 1 (Reference) | | 1 (Reference) | |
| | Aromatase inhibitor | 128 | 1.389 (0.971–1.985) | 0.072 | 1.231 (0.848–1.788) | 0.274 | 1.100 (0.742–1.631) | 0.634 | 1.135 (0.597–2.156) | 0.699 |
| | Tamoxifen | 297 | 1.467 (1.103–1.952) | 0.009 | 1.665 (1.244–2.227) | 0.001 | 1.612 (1.172–2.217) | 0.003 | 1.511 (0.843–2.708) | 0.165 |
| | **Fatty liver (+) at baseline** | | | | | | | | | |
| | Control | 70 | 1 (Reference) | | 1 (Reference) | | 1 (Reference) | | 1 (Reference) | |
| | Aromatase inhibitor | 71 | 1.514 (0.844–2.718) | 0.164 | 1.581 (0.87–2.871) | 0.133 | 1.749 (0.898–3.405) | 0.100 | 2.410 (0.666–8.722) | 0.180 |
| | Tamoxifen | 119 | 2.691 (1.61–4.497) | <0.001 | 2.672 (1.575–4.534) | <0.001 | 2.998 (1.646–5.462) | <0.001 | 3.668 (1.232–10.925) | 0.020 |
| **After matching** | **All** | | | | | | | | | |
| | Control | 256 | 1 (Reference) | | 1 (Reference) | | 1 (Reference) | | 1 (Reference) | |
| | Tamoxifen | 256 | 1.636 (1.237–2.165) | 0.001 | 1.714 (1.293–2.271) | <0.001 | 1.773 (1.294–2.43) | <0.001 | 1.716 (0.98–3.005) | 0.059 |
| | **Fatty liver (-) at baseline** | | | | | | | | | |
| | Control | 202 | 1 (Reference) | | 1 (Reference) | | 1 (Reference) | | 1 (Reference) | |
| | Tamoxifen | 186 | 1.481 (1.075–2.042) | 0.016 | 1.612 (1.166–2.23) | 0.004 | 1.590 (1.092–2.314) | 0.016 | 1.497 (0.745–3.007) | 0.257 |
| | **Fatty liver (+) at baseline** | | | | | | | | | |
| | Control | 54 | 1 (Reference) | | 1 (Reference) | | 1 (Reference) | | 1 (Reference) | |
| | Tamoxifen | 70 | 2.241 (1.244–4.038) | 0.007 | 2.219 (1.226–4.015) | 0.008 | 2.907 (1.451–5.821) | 0.003 | 279.154 (2.887–26992.851) | 0.016 |

Abbreviations: *HR*, hazard ratio; *CI*, confidence interval.

[*] Model 1: unadjusted.

[†] Model 2: adjusted for age, BMI, and underlying disease (diabetes, hypertension).

[§] Model 3: Model 2 plus cancer-related factors (stage, pathology, lymph node metastasis).

[¶] Model 4: Model 3 plus initial laboratory factors (FSH, platelet, AST, ALT, total bilirubin, total cholesterol, triglyceride, fasting blood glucose).

fatty liver from a large population. Since the use of tamoxifen has been extended to be 10 years, its side effect such as fatty liver is expected to increase even more.

The largest study of tamoxifen-induced fatty liver was conducted in 2005 enrolling 5,408 patients in Italy.[9] However, the major disadvantage of that study was that the occurrence of fatty liver was not evaluated by imaging study, but by an increase in liver enzyme. In addition, patients included in that study were not breast cancer patients, but healthy women who underwent hysterectomy. Thereafter, although several studies were reported, the patient group was so heterogeneous as to come to a solid conclusion, and factors that could affect fatty liver

**Table 6. Multivariable Cox proportional hazards regression for death.**

| Variable | Multivariable | |
|---|---|---|
| | HR (95% CI) | P-value |
| **All (N = 911)** | | |
| Fatty liver progression | | |
| No | 1 (Reference) | |
| Yes | 0.753 (0.380–1.493) | 0.417 |
| Age (year) | 1.041 (1.014–1.068) | 0.003 |
| ER (Intermediate or High) | 0.314 (0.164–0.599) | <0.001 |
| Cancer stage | | |
| ≤1 | 1 (Reference) | |
| 2 | 1.647 (0.795–3.412) | 0.180 |
| ≥3 | 3.789 (1.587–9.045) | 0.003 |

Abbreviations: ER, estrogen receptor; HR, hazard ratio; CI, confidence interval.

development were not adjusted.[19, 20] In this study, we presented the risk of tamoxifen *versus* control or AI accurately using PS matching and multiple Cox regression analysis with a large cohort of patients. Tamoxifen had a great effect both on the development of *de novo* fatty liver and fatty liver worsening than other treatment modalities. In addition, there was a steady increase in fatty liver incidence in proportion to the cumulative dose as seen on the Kaplan-Meier curve.

The mechanism of how tamoxifen causes fatty liver is not well understood yet. The most representative hypothesis is the "multiple hit" hypothesis which suggests that fat accumulation would make liver vulnerable to oxidants, and then second hits (*e.g.*, tamoxifen) could promote progress to steatohepatitis.[9] In this process, production of large amounts of reactive oxygen species, decreased mitochondrial β-oxidation, and increased TNF-α is known to play a major role in drug-induced steatohepatitis.[21] Our study supported this hypothesis with the fact that tamoxifen was more associated with deterioration of fatty liver in patients with pre-existing fatty liver than in those without.

The second hypothesis is that tamoxifen decreases lipid metabolism by inhibiting estrogen. Estrogen can regulate hepatic lipid metabolism through ERα, ERβ or GPER with regard to the extent of liver gene expression.[22] Indeed, the incidence of fatty liver gradually increases after menopause. It can be interpreted that with greater estrogen inhibition, more fatty liver will occur. Tamoxifen is known to cause higher levels of estrogen deprivation than AI. While AI blocks estrogen production in menopausal women who have already shown decreased estrogen production, tamoxifen blocks estrogen by binding to its receptor in the premenopausal status.[23, 24] Our study also demonstrated that the incidence of fatty liver was proportional to the degree of estrogen deprivation, which was highest in the tamoxifen group followed by the aromatase inhibitor and control group.

Because tamoxifen can induce or aggravate fatty liver mainly in susceptible patients, it is important to do close surveillance in advance, especially in high risk groups such as those with high BMI or presence of basal fatty liver. However, whether tamoxifen should be discontinued after the development of fatty liver is currently unclear. In our study, the occurrence of fatty liver did not affect the prognosis of breast cancer. Thus, we are not certain of the benefits and harms of suspending tamoxifen intake because it is difficult to adequately reflect the risk of cardiovascular disease during the five-year follow-up period. The guideline recommends weight reduction by 10%, rather than stopping the drug immediately.[21] Also, there was a

report showing that moderate-intensity exercise of more than 150-minutes a week could reduce the risk of fatty liver development.[25] If fatty liver does not improve using this method, suspending tamoxifen or changing it to AI may be considered.

Our study has several limitations. First, the main limitation is predominant from its retrospective nature. The patients were not randomly assigned to the modalities which meant that the choice of modalities might have been biased. Also, there were some subjects who had no imaging study during the follow-up period which resulted in missing data. Along with the above drawbacks, this study has been designed for bias-reduction by propensity score matching. This result can be used to implement further prospective cohort study. Second, the quantification of steatosis was evaluated by either USG or CT. For the diagnostic ability of non-enhanced CT for fatty liver, sensitivity of macrovesicular steatosis of 30% or greater was 0.991 (95% $CI$: 0.960–0.999), and specificity was 1.0 (95% $CI$: 0.97, 1.00).[26, 27] In general, diagnostic performance of non-enhanced CT is comparable to USG and MR-PDFF.[28] Considering accessibility and price, it has slight advantages over MRI, and the inter-reader and intra-reader accordance is superior to ultrasound. In addition, recent studies by Artz et al.[29] and Luca et al.[15] suggested that simple ROI-based mean attenuation measurement with CT has an inverse linear relationship with fat content measured by MR spectroscopy. These studies imply that CT may be used as a quantitative imaging biomarker for both detection and quantification of liver fat content. Third, we did not evaluate whether fatty liver was improved after discontinuation of tamoxifen because of the relatively short follow–up period. It would ascertain the causal relationship between tamoxifen use and fatty liver progression. Lastly, studies on Asians may not be applicable to all other races.

In conclusion, tamoxifen may be an important risk factor in the incidence and progression of fatty liver. Close follow-up and screening are necessary for high risk patients. Also, multidisciplinary approach with hepatologists should be considered for these populations. In the future, studies using accurate steatosis quantification (e.g., transient elastography, MR-based technique) may be needed with a large, prospective cohort.

## Supporting information

**S1 Fig. The flow chart of patients assessed for eligibility.**
(DOCX)

**S2 Fig. Distribution of propensity score for balance between groups and covariance balance plot.**
(DOCX)

**S3 Fig. Difference in survival rate according to aggravation of fatty liver.** (A) Before matching, (B) After matching.
(DOCX)

**S1 Table. Multivariable Cox proportional hazards regression for fatty liver aggravation (Before matching).**
(DOCX)

**S2 Table. Propensity score matching analysis.**
(DOCX)

**S3 Table. Univariable proportional hazards regression for fatty liver progression (After matching).**
(DOCX)

**S4 Table. Univariable proportional hazards regression for death.**
(DOCX)

**S5 Table. Propensity score matching analysis for the risk factors associated with fatty liver progression (caliper 0.3).**
(DOCX)

**S6 Table. Propensity score matching analysis for the risk factors associated with fatty liver progression (caliper 0.1).**
(DOCX)

## Acknowledgments

We thank Eun-Ae Jung (Librarian, Medical Library, Soonchunhyang University Bucheon Hospital) for carefully proofreading the manuscript.

## Author Contributions

**Conceptualization:** Jeong-Ju Yoo, Zisun Kim, Sang Gyune Kim.

**Formal analysis:** Jeong-Ju Yoo, Bora Lee.

**Investigation:** Yong Seok Lim, Min Sung Kim.

**Methodology:** Min Sung Kim, Bora Lee, Ji Eun Lee, Min Hee Lee.

**Supervision:** Bo-Yeon Kim, Young Seok Kim.

**Validation:** Bora Lee, Sang Gyune Kim.

**Visualization:** Bora Lee.

**Writing – original draft:** Jeong-Ju Yoo.

**Writing – review & editing:** Jeong-Ju Yoo, Sang Gyune Kim.

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
