## [Decision Letter · Decision Letter 0]

22 May 2020

PONE-D-20-05309

Risk of fatty liver after long-term use of tamoxifen in patients with breast cancer

PLOS ONE

Dear Dr. Kim,

Thank you for submitting your manuscript to PLOS ONE. After careful consideration, we feel that it has merit but does not fully meet PLOS ONE’s publication criteria as it currently stands. Therefore, we invite you to submit a revised version of the manuscript that addresses the points raised during the review process.

We look forward to receiving your revised manuscript.

Kind regards,

Hakan Buyukhatipoglu

PLOS ONE

Additional Editor Comments:

Authors spent a lot effort in this study and worths publishing. However it needs several improvements. Firstly please closely address the reviewers' suggestions. Additionally:

1) what kind of retrospective study is this) (ie. case-control, cohort etc.) You should include that. Which anova test did you use, one-way anova?

2) It seemed that a level of p<0.10 was used for selection of variables in order to include cox-regression analysis. You should explain about why you chose this significance level?

3) which model did you use for cox-regression (forward, enter, etc)?

4) Flow of discussion is not well. Especially first two paragraphs are not necessary looks like introduction. Please fully revise the discussion and discuss and focus on your objectives and results. Unnecessary review of tamoxifen is not appropriate.

5) Retrospective nature of this study is the main shortage of this study you should emphasize that in the limitations.

6) Please specifically address the statistical concerns of reviewer 2.

Journal Requirements:

2. In the ethics statement in the manuscript and in the online submission form, please provide additional information about the patient records used in your retrospective study, including:

a) whether all data were fully anonymized before you accessed them;

b) the date range (month and year) during which patients' medical records were accessed; and

c) the source of the medical records analyzed in this work (e.g. hospital, institution or medical center name).

Reviewers' comments:

Reviewer's Responses to Questions

**Comments to the Author**

1. Is the manuscript technically sound, and do the data support the conclusions?

Reviewer #1: Yes

Reviewer #2: Partly

Reviewer #3: Partly

2. Has the statistical analysis been performed appropriately and rigorously? 

Reviewer #1: Yes

Reviewer #2: I Don't Know

Reviewer #3: No

3. Have the authors made all data underlying the findings in their manuscript fully available?

Reviewer #1: No

Reviewer #2: No

Reviewer #3: Yes

4. Is the manuscript presented in an intelligible fashion and written in standard English?

Reviewer #1: Yes

Reviewer #2: Yes

Reviewer #3: Yes

5. Review Comments to the Author

Reviewer #1: This topic is of interest. The usage of tamoxifen is common in clinical work, therefore it is urgent to analyze the effect of tamoxifen on life quality and clinical prognosis in patients with breast cancer. In this retrospective study, the authors found that tamoxifen had a significant effect on the deterioration of fatty liver status compared to other treatment modalities in breast cancer patients. But fatty liver progression did not affect the prognosis of breast cancer patients and further follow-up might be needed and the effect of tamoxifen on cardiovascular disease needs further exploration. Here I have only several minor comments.

1.Statistical proper nouns such as 95%CI, IQR and HR should be expressed in italic form. The authors should check the manuscript and correct some erros.

2.The cross tab analysis was perfomed to compare the basline information between groups. However, some data meet the criteria for Fisher exact test in Table 1. The authors might check the data and methods and marked out the groups with Fisher exact test.

3.The list of abbreviations in page 2 was not in alphabetical order. I would suggest the author to rearrange in alphabetical order for reading convenience.

4.There are some typos and grammatical mistakes in this manuscript. But I believe that the author will check and correct in the revision.

Reviewer #2: It was interesting to read the paper ‘Risk of fatty liver after long-term use of tamoxifen in patients with breast cancer” by Yoo et al. where the authors highlight the aggravation of fatty liver disease by long term use of Tamoxifen by patients for breast cancer. It was a single-centred retrospective study.

The paper is well written but have raised a number if question which need further explanation. On balance, I feel that some of the important questions need explanation before considering this manuscript for publication. I have listed my comments as follows with my queries:

1- It’s a retrospective study which mean that the data collection was predominantly depended upon what was available from resources. This is a main drawback of most retrospective studies which result in difficulty of drawing any meaningful conclusion.

2- Ultrasonography is widely used for assessment and follow up of fatty liver disease however there are number of studies reporting significant interobserver and intra-observer variability in reporting the degree of fatty liver disease even by experienced radiologists. Such variabilities were also reported in term of CT scan evaluation of fatty liver disease. In this study, baseline assessment of fatty liver disease was done by ultrasonography and CT scan. The author stated that all UDSG and CT images were reported by two radiologists who were blinded to the study aims and drugs but the Autor didn’t explain whether all patients have ultrasound scans as well as CT or some have USG while others have CT? Also it is important to explain whether follow up assessment was done by the same mode of imaging as base line? Nevertheless, there remains a significant weakness in this study due to interobserver / intraobserver variability leading to lack of solid conclusion on progression of fatty liver disease. Some of the related referenced are as follows.

Ref:

Cengiz, Mustafa et al. “Sonographic assessment of fatty liver: intraobserver and interobserver variability.” International journal of clinical and experimental medicine vol. 7,12 5453-60. 15 Dec. 2014

Hamer OW, Aguirre DA, Casola G, Lavine JE, Woenckhaus M, Sirlin CB. Fatty liver: imaging patterns and pitfalls. Radiographics. 2006;26:1637–53.

3- The author didn’t explain whether the baseline assessment of fatty liver disease was done at the same post-operative intervals in all groups as there are studies reporting the presence of some degree of fatty changes in the liver post-operatively which improve with times.

4- It would be important to know the dietary habits of these patient as there are studies reporting association of fatty liver disease with diet.

5- Similarly, a detailed drug history is also important as patients in this study had underlying Diabetes and Hypertension but it is important to know the type of antihypertensive and diabetic medication in each group

6- It is also important to know whether any patient was taken off Tamoxifen during the follow up period and any subsequent improvement in the hepatic steatosis.

7- In discussion section, it should be mentioned about any correlation between degree of hepatic steatosis with drug dosage or duration. Also whether the changes in liver enzymes were in line with worsening of hepatic steatosis ?

Once authors have explained this queries , the manuscript can be resubmitted for further review .

Reviewer #3: A well-done, medium-volume, retrospective study. But no new data. My comments to the authors were as follow:

1- What is meant by “significant alcohol consumption” term at the exclusion criteria?

2- When you look at the values given at the table I; the difference between the groups for “age, BMI, platelets, AST, ALT, albümin, bilirubin, fasting blood glucose” looks too small. So p values seem to be wrong. Please check.

3- I cannot judge about the accuracy of propensity score matching analysis. I recommend an extra evaluation by a qualified statistician.

4- At table I what is meant by “treatment duration”; follow-up time?, chemotherapy?, tamoxifen. What is the treatment of control group?

5- No data was given about the chemotherapy? Chemotherapeutic agents can also cause fatty liver.

6- Along with ultrasonography, fibroscan can be a more valuable method of monitoring.

7- Some current and valuable articles are not referenced. Below you can fınd two sample:

a) Lee B, Jung EA, Yoo JJ, Kim SG, Lee CB, Kim YS, Jeong SW, Jang JY, Lee SH, Kim HS, Jun BG, Kim YD, Cheon GJ. Prevalence, incidence and risk factors of tamoxifen-related non-alcoholic fatty liver disease: A systematic review and meta-analysis. Liver Int. 2020 Mar 14. doi: 10.1111/liv.14434.

b) Hong N, Yoon HG, Seo DH, Park S, Kim SI, Sohn JH, Rhee Y. Different patterns in the risk of newly developed fatty liver and lipid changes with tamoxifen versus aromatase inhibitors in postmenopausal women with early breast cancer: A propensity score-matched cohort study. Eur J Cancer. 2017 Sep;82:103-114. doi: 10.1016/j.ejca.2017.05.002.

6. PLOS authors have the option to publish the peer review history of their article (what does this mean?). If published, this will include your full peer review and any attached files.

Reviewer #1: No

Reviewer #2: No

Reviewer #3: Yes: Huseyin Alkim

---

## [Author Response · Author response to Decision Letter 0]

4 Jul 2020

Response to the Associate Editor’s and Reviewers’ Comments

3 July, 2020

Dear reviewers and editorial staffs at Plos One, 

We are sincerely grateful for your thorough consideration and review of our manuscript, “Risk of Fatty Liver after Long-Term Use of Tamoxifen in Patients with Breast Cancer”, control number PONE-D-20-05309. Thanks to the accurate comments made by the reviewers, we could better understand the critical issues in this paper. We have modified the manuscript according to the reviewer’s suggestions. We hope that our revised manuscript will be considered and accepted for publication in Plos One. We acknowledge that the scientific and clinical quality of our manuscript has been improved with the guidance of the reviewers and editors.

The changes within the revised manuscript were highlighted (underlined and in blue). Point-by-point responses to the reviewers’ comments are provided below.

Editor:

<GENERAL COMMENTS>

1) Editor’s comment: What kind of retrospective study is this? (i.e. case-control, cohort etc.) You should include that. Which anova test did you use, one-way anova?

Author’s response: Thank you for the valuable advice. This study was a retrospective cohort study. We added this information to the Method section according to the editor’s recommendation. Regarding the anove test, we used one-way ANOVA test. We also added this information to the Method section. The Method of the manuscript was revised as follows:

“This was a single-center, retrospective cohort study.” (page 12, lines 14 –page 13, lines 2) 

“Statistical differences between groups were investigated using one-way analysis of variance (ANOVA) or Student’s t-test for continuous variables and chi-squares test or Fisher’s exact test for categorical variables as appropriate.” (page 12, lines 14 –page 13, lines 2) 

2) Editor’s comment: It seemed that a level of p<0.10 was used for selection of variables in order to include cox-regression analysis. You should explain about why you chose this significance level?

Author’s response: Thank you for the valuable advice. It is difficult to include all the variables into the multiple regression model due to limited number of events according to the practical thumb of rule, which is based on the following reference. Thus, we chose the criterion of 0.1 in our study. We described this fact in statistical analysis and added the reference. The Method section of the manuscript was revised as follows:

“Any variable showing a significance at 0.1 in the univariate model was selected as a candidate for the multivariable model [17].” (page 12, lines 14 –page 13, lines 2) 

“(Reference) 17. Hosmer Jr, D. W., Lemeshow, S., & May, S. (2011). Applied survival analysis: regression modeling of time-to-event data (Vol. 618). John Wiley & Sons.” (page 12, lines 14 –page 13, lines 2) 

3) Editor’s comment: Which model did you use for cox-regression (forward, enter, etc)?

Author’s response: Thank you for the valuable advice. We used the step-wise selection for the final multiple model. We added this method in the statistical analysis. The Method section of the manuscript was revised as follows:

“The final multiple Cox PH regression model was chosen by the stepwise selection based on the Akaike information criterion (AIC).” (page 12, lines 14 –page 13, lines 2) 

4) Editor’s comment: Flow of discussion is not well. Especially the first two paragraphs do not necessary look like the introduction. Please fully revise the discussion and discuss and focus on your objectives and results. Unnecessary review of tamoxifen is not appropriate.

Author’s response: Thank you for the valuable advice. As recommended, the two unnecessary paragraphs of the discussion were deleted, and the other paragraph of the discussion was modified. 

5) Editor’s comment: Retrospective nature of this study is the main shortage of this study you should emphasize that in the limitations.

Author’s response: Thank you for the valuable advice. For limitations arising from retrospective research, content was added to the discussion as recommended. The discussion of the manuscript was revised as follows:

“Our study has several limitations. First, the main limitation is predominant from its retrospective nature. The patients were not randomly assigned to the modalities which meant that the choice of modalities might have been biased.” (page 12, lines 14 –page 13, lines 2) 

6) Editor’s comment: Please specifically address the statistical concerns of reviewer 2.

Author’s response: Thank you for the advice. The detailed answers to reviewer 2's questions are listed below. 

Reviewer #1 :

<GENERAL COMMENTS>

1) Reviewer’s comment: Statistical proper nouns such as 95%CI, IQR and HR should be expressed in italic form. The authors should check the manuscript and correct some errors.

Author’s response: Thank you for the valuable advice. We corrected the above terms in italic form as recommended. 

2) Reviewer’s comment: The cross tab analysis was performed to compare the baseline information between groups. However, some data meet the criteria for Fisher exact test in Table 1. The authors might check the data and methods and marked out the groups with Fisher exact test.

Author’s response: Thank you for the valuable advice. We additionally indicated items using the Fisher exact test. We added a footnote on Table 2 and described the method in statistical analysis. It is worth noting that stage and pathology variables received warnings from the chi-squares test due to the low expected frequency, however the numerous cells of these variables prevented testings by the Fisher’s exact test. Thus, we used the chi-squares test in those cases. The Method section of the manuscript was revised as follows:

“Statistical differences between groups were investigated using one-way analysis of variance (ANOVA) or Student’s t-test for continuous variables and chi-squares test or Fisher’s exact test for categorical variables as appropriate.” (page 4, lines 19 –page 5, lines 3) 

3) Reviewer’s comment: The list of abbreviations in page 2 was not in alphabetical order. I would suggest the author to rearrange in alphabetical order for reading convenience.

Author’s response: Thank you for the valuable advice. We rearranged the abbreviations in alphabetical order.

4) Reviewer’s comment: There are some typos and grammatical mistakes in this manuscript. But I believe that the author will check and correct in the revision.

Author’s response: Thank you for the valuable advice. We received English proofreading once more and made the corrections.

Reviewer #2 :

<GENERAL COMMENTS>

1) Reviewer’s comment: It’s a retrospective study which mean that the data collection was predominantly depended upon what was available from resources. This is a main drawback of most retrospective studies which result in difficulty of drawing any meaningful conclusion.

Author’s response: Thank you for the valuable advice. For limitations arising from retrospective research, content was added to the Discussion section as recommended. The Discussion section of the manuscript was revised as follows:

“First, the main limitation is predominant from its retrospective nature. The patients were not randomly assigned to the modalities which meant that the choice of treatment might have been biased. Also, there were some subjects who had no imaging study during the follow-up period which resulted in missing data. Along with the above drawbacks, this study has been designed for bias-reduction by propensity score matching. This result can be used to implement further prospective cohort study.” (page 15, lines 13 –17) 

2) Reviewer’s comment: Ultrasonography is widely used for assessment and follow up of fatty liver disease however there are number of studies reporting significant interobserver and intra-observer variability in reporting the degree of fatty liver disease even by experienced radiologists. Such variabilities were also reported in term of CT scan evaluation of fatty liver disease. In this study, baseline assessment of fatty liver disease was done by ultrasonography and CT scan. The author stated that all UDSG and CT images were reported by two radiologists who were blinded to the study aims and drugs but the Author didn’t explain whether all patients have ultrasound scans as well as CT or some have USG while others have CT? Also it is important to explain whether follow up assessment was done by the same mode of imaging as base line? Nevertheless, there remains a significant weakness in this study due to interobserver / intraobserver variability leading to lack of solid conclusion on progression of fatty liver disease. Some of the related referenced are as follows.

Ref:

Cengiz, Mustafa et al. “Sonographic assessment of fatty liver: intraobserver and interobserver variability.” International journal of clinical and experimental medicine vol. 7,12 5453-60. 15 Dec. 2014

Hamer OW, Aguirre DA, Casola G, Lavine JE, Woenckhaus M, Sirlin CB. Fatty liver: imaging patterns and pitfalls. Radiographics. 2006;26:1637–53.

Author’s response: Thank you for the valuable advice. We used only one technique (among CT or USG) in a patient as a diagnostic tool to determine fatty liver thoughout the follow-up period. For example, if a patient was first diagnosed fatty liver by ultrasound, the change of fatty liver was then determined by ultrasound. We added this information in the Method section as follows: 

“We used only one technique (among CT or USG) as a diagnostic tool to determine fatty liver thoughout the follow-up period.” (page 15, lines 8 –9) 

We acknowledge that what Reviewer #2’s pointed out is the main limitation of our research. CT and USG are most common modalities used for follow-up of breast cancer, as is recommended in the NCCN guideline.1. For the diagnostic ability of non-enhanced CT for fatty liver, sensivitiy of macrovesicular steatosis of 30% or greater was 0.991 (95% confidence interval: 0.960, 0.999), and specificity was 100% (95%CI: 97-100).2,3 In general, diagnostic performance of non-enhanced CT is comparable to USG and MR-PDFF.4. However, considering accessibility and price, it has advantages over MRI, and the inter-reader and intra-reader accordance is superior to ultrasound. In addition, recent studies by Artz et al.5 and Luca et al.6 suggested that simple ROI-based mean attenuation measurement with CT has an inverse linear relationship with fat content measured by MR spectroscopy. These studies imply that CT may be used as a quantitative imaging biomarker for both detection and quantification of liver fat content. We added this infomation in the Discussion section. The Discussion of the manuscript was revised as follows:

“Second, the quantification of steatosis was evaluated by either USG or CT. For the diagnostic ability of non-enhanced CT for fatty liver, sensitivity of macrovesicular steatosis of 30% or greater was 0.991 (95% CI: 0.960-0.999), and specificity was 1.0 (95% CI: 0.97, 1.00). In general, diagnostic performance of non-enhanced CT is comparable to USG and MR-PDFF. Considering accessibility and price, it has slight advantages over MRI, and the inter-reader and intra-reader accordance is superior to ultrasound. In addition, recent studies by Artz et al. and Luca et al. suggested that simple ROI-based mean attenuation measurement with CT has an inverse linear relationship with fat content measured by MR spectroscopy. These studies imply that CT may be used as a quantitative imaging biomarker for both detection and quantification of liver fat content.” (page 15, lines 8 –9) 

3) Reviewer’s comment: The author didn’t explain whether the baseline assessment of fatty liver disease was done at the same post-operative intervals in all groups as there are studies reporting the presence of some degree of fatty changes in the liver post-operatively which improve with time.

Author’s response: Thank you for the valuable advice. Follow-up CT and ultrasound were performed every 6months for patients with high risk of recurrence, and every 12months for other patients, as recommended in the NCCN guideline.1. We added this infomation in the Method section which was revised as follows: 

“To monitor possible recurrences of breast cancer, follow-up computed tomography (CT) or ultrasonography (USG) were performed every 6months for patients with high risk of recurrence, and every 12months for others as recommended in the NCCN guideline.” (page 12, lines 14 –page 13, lines 2) 

4) Reviewer’s comment: It would be important to know the dietary habits of these patient as there are studies reporting association of fatty liver disease with diet.

Author’s response: Thank you for the valuable advice. In this study, the data of dietary habits were insufficient in the medical charts and could not be investigated. We are currently planning a prospective study and will definitely consider this point in the next one. 

5) Reviewer’s comment: Similarly, a detailed drug history is also important as patients in this study had underlying Diabetes and Hypertension but it is important to know the type of antihypertensive and diabetic medication in each group. 

Author’s response: Thank you for the valuable advice. In this study, medication intake was investigated by group, but information on individual drugs was insufficient and could not be investigated. We will definitely consider this in the subsequent study.

6) Reviewer’s comment: It is also important to know whether any patient was taken off Tamoxifen during the follow up period and any subsequent improvement in the hepatic steatosis.

Author’s response: Thank you for the valuable advice. The study did not include patients who discontinued tamoxifen. We agree that in the long run, changes after tamoxifen discontinuation are also essential. We added this limitation in the Discussion section. 

“Third, we did not evaluate whether fatty liver was improved after discontinuation of tamoxifen because of the relatively short follow–up period. It would ascertain the causal relationship between tamoxifen use and fatty liver progression.” (page 12, lines 14 –page 13, lines 2) 

7) Reviewer’s comment: In discussion section, it should be mentioned about any correlation between degree of hepatic steatosis with drug dosage or duration. Also whether the changes in liver enzymes were in line with worsening of hepatic steatosis?

Author’s response: Thank you for the valuable advice. We analyzed the association between the severity of fatty liver at baseline with the drug duration and ALT at baseline using the spearman’s correlation coefficient. They have no significant or clinically meaningful correlation as shown in following figures. We added this information in the Result section. 

“Next, we performed a subgroup analysis in the tamoxifen group to find any correlations between the degree of hepatic steatosis with drug duration or changes in liver enzymes. However, we discovered no significant or clinically meaningful correlations when using the spearman’s correlation coefficient.” (page 12, lines 14 –page 13, lines 2) 

Reviewer #3 :

<GENERAL COMMENTS>

1) Reviewer’s comment: What is meant by “significant alcohol consumption” term at the exclusion criteria?

Author’s response: Thank you for the valuable advice. We used the AASLD guideline and criteria. Ongoing or recent alcohol consumption >21 standard drinks on average per week in men and >14 standard drinks on average per week in women is used as a reasonable threshold for significant alcohol consumption. We added this information to the Method section which was revised as follows:

“Ongoing or recent alcohol consumption >21 standard drinks on average per week in men and >14 standard drinks on average per week in women was the criteria for significant alcohol consumption.” (page 15, lines 13 –17) 

2) Reviewer’s comment: When you look at the values given at the table I; the difference between the groups for “age, BMI, platelets, AST, ALT, albümin, bilirubin, fasting blood glucose” looks too small. So p values seem to be wrong. Please check.

Author’s response: According to your considerable comment, we checked all the statistical results including Table 1, and they turned out to be identical to the previous ones. The difference of age, BMI, platelets, AST, ALT, albumin, bilirubin and fasting blood glucose seem to be small, but the large sample size might be beneficial to show the statistical significance since they have the small standard errors. 

3) Reviewer’s comment: I cannot judge about the accuracy of propensity score matching analysis. I recommend an extra evaluation by a qualified statistician.

Author’s response: Thank you for the considerable comment. The caliper of 0.05 was applied in this study, but we generated two more matched sets assuming the narrower caliper, 0.03 and 0.01 for validation of propensity score matching. From those sets, we obtained the similar pattern of results related to the fatty liver progression of tamoxifen and added the supplementary tables 5 and 6. Also, we described the additional analysis for this validation in statistical analysis and result. The Method and Result section of the manuscript was revised as follows:

“We used the nearest available matching (1:1) method for PS matching with the caliper of 0.05. For validation , we generated additional matched datasets using the caliper of 0.03 and 0.01.” (page 15, lines 13 –17) 

“The effect of tamoxifen on fatty liver progression was also confirmed to the same degree and significance in other matched datasets (HR: 1.4, 95% CI: 1.018-1.927 in dataset with caliper 0.3; HR: 1.452, 95% CI: 1.021-2.065 in dataset with caliper 0.1) (Supplementary Table 5 and 6).” (page 15, lines 13 –17) 

4) Reviewer’s comment: At table I what is meant by “treatment duration”; follow-up time?, chemotherapy?, tamoxifen. What is the treatment of control group?

Author’s response: The definition of "treatment duration" is the duration of taking tamoxifen or aromatase inhibitor, respectively, in the tamoxifen or AI groups. In the case of the control group, it means the period of receiving chemotherapy or other conservative management.

5) Reviewer’s comment: No data was given about the chemotherapy? Chemotherapeutic agents can also cause fatty liver.

Author’s response: The chemotherapeutic agents of anticancer drugs are so diverse that they could not be described individually, but mostly anthracycline and taxane agents were used. In the case of the anti-cancer agent described above, there were no reports of significant fatty liver. The Method section of the manuscript was revised as follows:

“The chemotherapeutic agents of anticancer drugs were diverse, however, anthracycline and taxane-based chemotherapy was most widely used. ” (page 15, lines 13 –17) 

6) Reviewer’s comment: Along with ultrasonography, fibroscan can be a more valuable method of monitoring.

Author’s response: Thank you for the valuable advice. We totally agree with the reviewer’s opinion. The proportion of patients who received fibroscan was small and could not be analyzed in this study. We are currently planning a prospective study and will definitely consider it in the next one.

7) Reviewer’s comment: Some current and valuable articles are not referenced. Below you can fınd two samples:

a) Lee B, Jung EA, Yoo JJ, Kim SG, Lee CB, Kim YS, Jeong SW, Jang JY, Lee SH, Kim HS, Jun BG, Kim YD, Cheon GJ. Prevalence, incidence and risk factors of tamoxifen-related non-alcoholic fatty liver disease: A systematic review and meta-analysis. Liver Int. 2020 Mar 14. doi: 10.1111/liv.14434.

b) Hong N, Yoon HG, Seo DH, Park S, Kim SI, Sohn JH, Rhee Y. Different patterns in the risk of newly developed fatty liver and lipid changes with tamoxifen versus aromatase inhibitors in postmenopausal women with early breast cancer: A propensity score-matched cohort study. Eur J Cancer. 2017 Sep;82:103-114. doi: 10.1016/j.ejca.2017.05.002.

Author’s response: Thank you for the valuable advice. In the case of the first paper, it is a paper written by our group. We are honored to have it recommended and added it directly to the list of references.

---

## [Editor Report · Decision Letter 1]

9 Jul 2020

Risk of fatty liver after long-term use of tamoxifen in patients with breast cancer

PONE-D-20-05309R1

Dear Dr. Kim,

We’re pleased to inform you that your manuscript has been judged scientifically suitable for publication and will be formally accepted for publication once it meets all outstanding technical requirements.

Kind regards,

Academic Editor

PLOS ONE
---

## [Editor Report · Acceptance letter]

15 Jul 2020

PONE-D-20-05309R1 

Risk of fatty liver after long-term use of tamoxifen in patients with breast cancer 

Dear Dr. Kim:

I'm pleased to inform you that your manuscript has been deemed suitable for publication in PLOS ONE. Congratulations! Your manuscript is now with our production department. 

Kind regards, 

on behalf of

Dr. Hakan Buyukhatipoglu 

Academic Editor

PLOS ONE